# Piton: Investigating the Controllability of a Wearable Telexistence Robot

**DOI:** 10.3390/s22218574

**Published:** 2022-11-07

**Authors:** Abdullah Iskandar, Mohammed Al-Sada, Tamon Miyake, Yamen Saraiji, Osama Halabi, Tatsuo Nakajima

**Affiliations:** 1Department of Computer Science and Engineering, Waseda University, Tokyo 169-8555, Japan; 2Department of Computer Science and Engineering, Qatar University, Doha 2713, Qatar; 3Future Robotics Organization, Waseda University, Tokyo 169-8555, Japan; 4Avatarin Inc., Tokyo 103-0022, Japan

**Keywords:** wearable robot, appendage, snake-shaped, serpentine robot, telepresence robot, telexistence, communication, teleoperation

## Abstract

The COVID-19 pandemic impacted collaborative activities, travel, and physical contact, increasing the demand for real-time interactions with remote environments. However, the existing remote communication solutions provide limited interactions and do not convey a high sense of presence within a remote environment. Therefore, we propose a snake-shaped wearable telexistence robot, called Piton, that can be remotely used for a variety of collaborative applications. To the best of our knowledge, Piton is the first snake-shaped wearable telexistence robot. We explain the implementation of Piton, its control architecture, and discuss how Piton can be deployed in a variety of contexts. We implemented three control methods to control Piton: HM—using a head-mounted display (HMD), HH—using an HMD and hand-held tracker, and FM—using an HMD and a foot-mounted tracker. We conducted a user study to investigate the applicability of the proposed control methods for telexistence, focusing on body ownership (Alpha IVBO), mental and physical load (NASA-TLX), motion sickness (VRSQ), and a questionnaire to measure user impressions. The results show that both the HM and HH provide relevantly high levels of body ownership, had high perceived accuracy, and were highly favored, whereas the FM control method yielded the lowest body ownership effect and was least favored. We discuss the results and highlight the advantages and shortcomings of the control methods with respect to various potential application contexts. Based on our design and evaluation of Piton, we extracted a number of insights and future research directions to deepen our investigation and realization of wearable telexistence robots.

## 1. Introduction

### 1.1. Background

With the emergence of the COVID-19 pandemic in 2020, global restrictions were imposed on various sectors and activities, including overseas travel, close physical contact, as well as training and educational activities at physical facilities. The pandemic highlighted the importance of video conferencing and telecommunication within both industrial and educational and academic contexts. Various video conferencing solutions have advanced communication and collaborative work innovations that attempt to replace close and physical contact [1].

Telepresence is an emerging medium for communication, where instruments such as video cameras and microphones are intermediaries for the participant’s senses [2]. Telepresence has been used in a variety of novel application domains, such as office settings [3,4], education [5], and medical applications [6]. However, telepresence systems have drawbacks: most systems do not enable a high sense of immersion, body ownership, or enable physical interaction. In contrast, telexistence refers to a group of technologies aiming at conveying high sense presence within remote environments through various modalities, including visual, auditory, and real-time physical interactions [7,8]. Despite the potential of telexistence systems, the amount of research that focuses on telexistence remains limited.

Most telexistence systems have demonstrated robots that focus on anthropomorphic (human-like) implementations, which are limited as they only provide positional and rotational movements similar to the human head. Generally, having a large robot workspace and flexible movement are essential for object inspection and interaction within daily life and industrial contexts [9,10]. Accordingly, this research presents a novel wearable telexistence robot, “Piton”, which is designed based on a snake-like structure with high degrees of freedom (DoFs) and redundancy. Piton is worn by a user (surrogate), and it is operated by another user at a different location. Piton provides auditory feedback, stereoscopic visual feedback, and rotational and positional movements of the camera system in a snake-shaped form-factor for a remote-user. In addition, Piton’s snake form-factor enables parallax motion, inspection of objects from multiple angles, and head gestures which can be used within social contexts. Our implementation consists of a local system and a remote system. The local system includes a head-mounted display (HMD), a motion-tracking system, and an interface for initiating various system functions. The remote system consists of a wearable snake-shaped robot with a stereo camera as an end-effector, a robot control system, and an audio-visual communication system.

Although various works have proposed different control methods for wearable robots [11,12,13], these control methods focus on exoskeletons and rehabilitation tasks, which are completely different and have different design objectives than telexistence robots. Accordingly, previous works in telexistence have proposed various control methods, such as using HMDs [14,15,16] and foot-mounted trackers [17,18] to remotely control robots. Accordingly, we extend these previous works to implement and explore the controllability of Piton by implementing the following control methods: (1) using the HMD to control the position and orientation of Piton (HM); (2) using the HMD to control the rotation and a hand-held tracker to control the position of Piton (HH); and (3) using the HMD to control the rotation and a foot-mounted tracker to control the position of Piton (FM).

Telexistence systems have several critical requirements for enabling a highly immersive telexistence experience. These requirements include low mental workload [17,19], low motion sickness effects [20], and high body ownership over the remote system while interacting with the remote environment [21]. Accordingly, we focused our evaluation objective on gauging and comparing the suitability of Piton’s three control methods for use in telexistence experiences. Therefore, we used the Task Load Index (NASA-TLX) [22] to measure mental and physical workloads, the Virtual Reality Sickness Questionnaire (VRSQ) [23] to measure motion sickness effects, and the Illusion Virtual Body Ownership questionnaire (Alpha IVBO) [24].

We hired 17 subjects to evaluate three control methods to do three tasks: mirroring on a screen (Task 1), finding letters and numbers on Uno Stacko (Task 2), and reading a text on wrapped cardboard (Task 3). These three tasks are executed using the three control methods. Overall, the results indicate that Piton can be used for telexistence, as all three control methods demonstrated overall good scores in the mentioned questionnaires. Such findings are critical, since this is the first study to provide evidence that snake-shaped robotic systems can be used for telexistence, which paves the way to investigate novel robotic form-factors or control methods for use in telexistence.

This paper consists of ten sections, commencing with the first section for the introduction. Section 2 explains the related work. Section 3 introduces the design concept. Section 4 describes Piton’s novel interactions and contexts of use. Section 5 explains the system implementation. Section 6 describes the Piton control methods and their calibration process. Section 7 describes the evaluation and results. Section 8 provides a discussion of the quantitative and qualitative results. Section 9 describes the future work and conclusions of the paper.

### 1.2. Contribution

We summarize the contributions of our paper as follows:

1.To the best of our knowledge, our work is the first to design, fabricate, and explore using a wearable snake-shaped robot for telexistence. Although
previous works have explored the snake form-factor for various domains, such as
for a wearable multipurpose appendage or for multi-haptic feedback in VR [9,25], the snake form-factor was not evaluated in
any previous work within the telexistence research domain. Our literature
survey shows that telexistence robots, whether wearable or not, are mainly implemented
as anthropomorphic robots; the shape of the robot’s head and structure
replicates that of a human. This contribution is significant, as nonhuman telexistence
robots are unexplored within the domain of telexistence. Moreover, we show that
a snake-like robotic system can be advantageous in various scenarios when
compared to human-like telexistence systems. For example, we demonstrate how
Piton can use its malleable body to inspect objects from multiple points of
view, and its large workspace can be used to gain better awareness of the
surrounding environment.2.We present three control methods for using Piton while complying with
the requirements of telexistence. Previous efforts in telexistence mainly focused on standard control methods that mimicked simple human head movements,
relying mainly on linear and direct mapping between the user’s head rotations
and those of the robot [15,16,26]. Unlike
previous works, our developed control methods utilize the user’s head, hands,
and feet while using linear and scalar mappings to control a snake-shaped
robot. This contribution advances the state-of-the-art design and development as
it proposes and evaluates novel control methods for use in telexistence robots
and proceeds to provide evidence of their validity for telexistence
(contribution 3).3.We provide user study results which indicate that Piton can be used for
telexistence. We evaluated the three developed control methods against the main
requirements of telexistence, which are a high sense of body ownership, low
motion sickness, and low mental and physical load. The results show that the
three control methods generally meet telexistence requirements. This
contribution is significant for telexistence, as the results and statistical
analysis show that a snake-like robot can be used for telexistence.

Overall, the contributions of this paper are critical to advance the state-of-the-art in telexistence robots. To the best of our knowledge, this work is the first to propose a snake-like telexistence robot and to propose three control methods for controlling the robot. Most significantly, the user study results of these three control methods show that Piton can be used for telexistence, as it meets the main requirements of telexistence systems. Therefore, such contributions are essential for paving the way for future generations of nonanthropomorphic telexistence robots that span beyond human-like robots and typical control methods.

## 2. Related Work

Our research expands on three main research areas: (1) nonrobotic telepresence systems, (2) robotic telepresence systems, and (3) telexistence systems. We explain each of these below.

A variety of researchers have explored nonrobotic telepresence systems. These systems mainly communicate primary modalities such as vision and auditory information. “Livemask” [27] is a facial screen that stands on a table as a monitor and which mimics a remote-side user’s face. The screen is considered a surrogate telepresence system that shows faces on a local screen based on a remote user’s 3D face data. Similarly, “ChameleonMask” [28] is a wearable screen worn by a surrogate user, where the screen shows a remote user’s face. The surrogate wearing the screen responds to requests from the remote user displayed on the wearable screen. “Livesphere” [29] is a head-mounted device, worn by a remote user, with six cameras on top of it to capture the surrounding area. The local user uses an HMD to see spherical images and control the movement via an HMD’s head rotation data.

Robotic telepresence systems extend the nonrobotic telepresence systems by providing higher movement flexibility and interactivity within the remote environment. TEROOS [30] is a wearable, small telepresence humanoid robot mounted on the user’s shoulder and controlled remotely by another user. This wearable robot can rotate 92 degrees horizontally and 110 degrees vertically, thereby enabling the controlling user to inspect remote environments. “WithYou” [31] is a wearable device with a pan-and-tilt camera and various sensors mounted on the remote user’s chest, allowing local users to use an HMD to control the pan and tilt motions of the camera, which is worn by a remote user.

Telexistence is a similar concept to telepresence, however, telexistence focuses on the highly-realistic sensation of existence in a remote location by engaging multiple sensory modalities [32]. TELESAR [7] is a telexistence robot, consisting of a humanoid robot and a cockpit system for controlling the robot. The cockpit consists of various control and feedback devices, including an HMD, speakers, a microphone, a haptic glove, and a motion tracking system. The remote location includes a humanoid robot with a stereoscopic camera, microphone, and speakers. The user controls a humanoid robot using motion tracking system, with markers on their head, shoulder, arm, hand, foot. Fusion [16] is a wearable telexistence system, where the local system consists of HMD(the Oculus CV1) for viewing and for head motion control, as well as a hand controllers for controlling the hands of the wearable robot. The remote system consists of a wearable backpack-like robot, comprising robotic head with stereoscopic vision, speakers, microphones, and two robotic arms (six DoFs) and 5-finger robotic hands. Al-Remaihi et al. [33] investigated a telexistence robotic system for remote physical object manipulation. The system consists of a robot arm with a gripper end-effector that can be remotely operated using an HMD and hand exoskeleton. The user at a local site controls the robot using a Vive Tracker [34] mounted on the hand, while the HMD is used to monitor the remote environment. A 3D-printed exoskeleton is used for both controlling the robotic-gripper and for receiving haptic feedback.

Unlike the majority of telepresence systems which do not exhibit a high sense of immersion, body ownership, or physical interactions, Piton advances the state-of-the-art development of telexistence systems. Piton is the first to utilize a wearable robot with a snake form-factor, which provides higher flexibility in movement to explore remote environments beyond previous systems. Piton’s flexibility allows for novel interactions that have not been explored before, such as inspecting and assisting in industrial or daily application contexts.

## 3. Design Concept

Our main objective is to design a snake-shaped, wearable telexistence robot sharing travel and work-related experiences with remote users. In order to develop a telexistence system for these purposes, we set several main design considerations and show how we satisfy them in Piton, in a similar fashion to previously developed robots [9,15,25,35,36]. Accordingly, our concept design has three primary design considerations: (1) telexistence, (2) flexible head movement, and (3) wearability. Below, we discuss how we aim to achieve each design consideration:Telexistence: A telexistence system requires low-latency stereoscopic visual feedback [37], low-latency auditory communication [38], and physical interactions with the remote environment [39].Flexible Head Movement: The robot should have flexible head movements with high DoFs and redundancy, similar to snake-shaped robots [25]. Such high-flexibility enable users to have large operational workspaces, which accordingly enable efficient inspection of remote environments and objects.Wearability: Our system should be designed as a wearable system so remote users can easily wear it and go to various locations. Wearable systems can also provide numerous interaction potentials to assist remote users in various contexts that are otherwise difficult to access through mobile robots (e.g., going up stairs or in narrow corridors).

We implemented the design considerations of Piton in a wearable robot system, comprising a local site (Figure 1a), where a user controls and communicates through the remote robot, and a remote site (Figure 1b), where a surrogate user wears the robot. The Appendix A show how the local and remote systems are implemented. We explain how we implemented the design considerations in our system as follows:
Telexistence: Our system uses a stereo camera system [40] embedded on the end-effector of the robot; auditory communication, including a mic and speaker (both explained in Section 5.2.2 and Section 5.3.2) on the local and remote system; and an HMD on the local system to enable an immersive experience (detailed in Section 5.2.1).Flexible Head Movement: The telexistence robot is snake-shaped with eight interlinked servomotors (explained in Section 5.3.1). Snake-type systems enable a high degree of redundancy, which in turn, allows the robot to be situated in a wide variety of postures using various movement trajectories.Wearability: We developed a lightweight robot system capable of being worn on various locations around the body, such as the shoulder or waist. The robot is mounted on a rack that is worn as a backpack, which enables users to conveniently wear or take-off the robot (explained in Section 5.3.1).

The term “Wearable robot” is mostly associated with traditional robotic systems, such as exoskeletons or rehabilitation robots [41]. However, novel forms of wearable robots emerged in recent years, where such robots are also identified as wearable robots. Such emerging wearable robots are designed to be continuously worn and used while worn, meeting the same presumptions of wearability of wearable devices and robots [42]. Some examples include supernumerary robotic limbs (SRLs) [9,35,43], wearable companion robots [44], haptic feedback robots [25], and telexistence wearable robots [16]. Accordingly, Piton is the first wearable snake-shaped telexistence robot that is designed to be ergonomically worn like a backpack, and it is designed to be used while being worn.

## 4. Piton’s Novel Interactions and Contexts of Use

Previous works show that the malleability of the snake form-factor enables robots to be flexible enough to be used in a large variety of usage contexts [9,10,25,45]. Similarly, Piton can be used in different unexplored interaction contexts, whether for daily use or industrial contexts. Piton can be used in daily usage contexts to enable companionship with a remote user [10]. For example, Piton can be used for interacting with the surrogate user (as shown in Figure 2a) or the remote environment, such as checking merchandise or enjoying activities with the surrogate (as shown in Figure 2b).

Piton can also be used within social contexts (e.g., social gatherings or parties), where it can engage in interactions with various users at the remote site (Figure 2c). Since Piton is a wearable robot, it can easily be taken to various locations, such as outdoors for hiking, shopping, or going to museums, which paves the way for various potential deployment contexts [46]. Moreover, Piton can also be used for practical use cases, such as guiding users at home to fix equipment or as a replacement for existing teleconferencing solutions [9].

Within industrial contexts, Piton can have a variety of advantages over existing systems since it can move and inspect objects from various directions and distances and can be used for skill-transfer applications (as shown in Figure 3a). For example, Piton can be deployed at a remote industrial location to inspect equipment or instruct workers on how to operate machinery correctly at distant locations (e.g., power plants, offshore oil rigs, different cities, etc.). In such scenarios, an expert controls Piton and carries out inspection tasks for environments or objects with a surrogate user (as shown in Figure 3b). The combination of telexistence, multimodal communication capabilities, and Piton’s large workspace enables it to offer a versatile user experience within remote environments, thereby potentially contributing to saving effort and time through remote work within various application contexts.

Overall, Piton can be used as a test-bed to explore the mentioned application contexts), which can pave the way for future implementations of Piton that focus on specific application domains. For example, within industrial tasks, Piton can be integrated with thermal imaging cameras or additional sensors. For daily usage contexts, Piton can be designed with smaller, slimmer, or unobtrusive form-factors to satisfy the requirements of daily use [9]. The Appendix A demonstrates a variety of novel potential application contexts of Piton.

## 5. System Implementation

Our implementation of Piton consists of two parts, a local system and a remote system. In the below subsections, we start by highlighting the systems integration architecture, followed by a detailed implementation of each of Piton’s systems at the remote and local sites.

### 5.1. System Integration

Our implementation consists of a system at the local site (local system) and a remote site (remote system), shown in Figure 4. This architecture extends common telepresence and telexistence systems architectures [28,46] by focusing on various control methods, an inverse kinematic model, and a novel robotic form-factor. The next subsections explain the local and remote system implementations in detail.

### 5.2. Local System

The local system consists of two main components: (1) the head mount display and tracking system and (2) the control system. Both main components are explained in the following section.

#### 5.2.1. HMD and Tracking System

We used the HTC Vive HMD, trackers and HTC Vive tracking system to track the user’s head orientation and to carry out positional control on the robot [34]. We evaluated three control methods (HM shown in Figure 5a, HH shown in Figure 5b, and FM is shown in Figure 5c). While HM is implemented only using the HMD, enabling both rotational and positional controls of Piton, two HTC Vive trackers were used to implement the foot and hand controls of the robot (further explained in Section 6). Moreover, the HMD is equipped with a built-in microphone and headphones to establish auditory communication with the remote user.

#### 5.2.2. Control System

We used the Unity3D engine [47] to develop the robot control system, integrate the stereoscopic video stream, and establish auditory communication. We chose Unity3D as it enabled us to easily integrate various control and feedback components to interface with the robot, HMD, its tracking system and interface with various hardware.

Robot Control: We implemented an inverse kinematic solver (IK) based on BioIK [48] so the robot IK solver can provide a solution for each of the robot’s servomotor. We used two IK solvers, one to set the rotations of Piton’s head (using the HMD in all the control methods), while the other was used to set the position of Piton (dependent on the chosen control method). For positional movements, we created a positional movement objective that can be moved using the three control methods (shown as Figure 6a,b). When the objective is moved, the IK solver attempts to find a solution that satisfies the positional objective, while also satisfying other constraints (keeping the robot’s head straight, angular limits., etc.). The IK solver was optimized to fulfill the movement objectives based on the three control methods needed for the user study (further discussed in Section 6.1). For example, if the objective is moved to the top or right positions (Figure 6a, Figure 6b) the robot’s model moves to the top or right position following the objective. The positional movement objective of the IK [48] is set for the fourth servomotor, so that we can obtain a solution for the first four servomotors of the robot. The IK solver is only used for positional movements, whereas rotational movements are read directly from the HMD (Further details in Section 6).

Accordingly, users only need to move the HMD (in HM control method) or HMD with trackers to control Piton (in HH and FM control methods). The HMD’s orientation data (based on IMU) and trackers’ positional data are read as inputs, where they are mapped to control the robot head’s rotations and the IK target objectives’ positions, respectively (further explained in Section 6. Piton Control Methods). When the user moves the HMD or trackers, the IK objectives associated with the HMD and trackers will also move, causing the IK system to recalculate a new solution to direct the robot movements (more details in Section 6.1). The angles from the IK system are read in real time and transmitted to the robot through WebSocket [9,49], after which they are executed. We also developed a simple interface to allows users to initiate and set-up various system functions for the robot, camera, and calibration system (Figure 7).

Our initial evaluation of latency, based on the network latency method [50,51] indicates an average of 11.64 ms (SD = 3.40), which was measured from the moment positional and rotational data of the IK model were read in the Unity3D environment to the moment the servomotor angles are executed on Piton. Previous works in telexistence and virtual reality indicate that latency should be less than 200 ms in order to provide a smooth experience and not induce simulator sickness [52,53]. Accordingly, we believe the latency of our system is within the acceptable range.

Stereoscopic Video Stream: Our system used the Gstreamer plugin for Unity3D, which was used and evaluated in previous work [39,54,55]. Once the camera video feed is received at the local system, it is divided into two images to be viewed by the left and right eyes in the HMD. Since our system replicates the same hardware and software implementation of a previous system [54], the estimated latency is 80 s, which was measured from when the ZED Mini Stereo Camera captures the stereo images at the remote site to the moment when the images are shown on the HMD at the local site.

Audio Streaming: We used the WebRTC Unity3D plug-in [56] to provide an easy and reliable audio communication method across the internet.

### 5.3. Remote System

The remote system comprises two main components: (1) the implementation of the Piton robot and (2) the robot controller software and audio-visual streaming. These components are explained in the following subsections.

#### 5.3.1. Piton Implementation

The structure of the robot consists of eight servomotors (eight DoFs) interlinked together in a way that is similar to snake-shaped robots. The robot was implemented using two Dynamixel MX64-AT servomotors, one MX106 servomotor, one 2XL430 servomotor, and three AX12A servomotors [57]. We slightly optimized the PID parameters for the experiment to reduce shaking and overshooting. The values for the first servomotor are (800, 100, 10,652); the second servomotor, (500, 100, 6000); and the third servomotor, (500, 100, 6000), whereas the rest of the motors use the manufacturer’s set PID parameters. The links between the servomotors were made of aluminum, whereas the last four servomotors were linked using PLA frames (Figure 8). The end-effector was custom-designed, and 3D printed with PLA to house the ZED Mini Camera [40]. The Appendix A further shows the robot from various angles and when being used.

Piton is mounted on a lightweight backpack rack and mainly constructed with aluminum (front view as shown in Figure 9a and side view as shown in Figure 9b). We designed the base motor’s enclosure, which was 3D printed using ABS, to be fitted on the user’s shoulder. The high DoF of the robot provides a flexible workspace for moving and situating the end-effector in different postures to inspect remote objects.

The total weight of the robot is 0.98 kg, and its length is 53 cm. The weight of a single-board computer (LattePanda Alpha) is 0.35 kg, the backpack rack is 1.05 kg, the microphone is 0.07 kg, the speaker is 0.03 kg, and the Zed Mini Camera is 0.06 kg. Therefore, the total weight of the Piton is approx. 2.54 kg.

#### 5.3.2. Robot Control Software and Audio-Visual Streaming

Robot Control: The robot was connected to LattePanda Alpha [58], which is a low-powered PC. Our robot control software extends previous works [9,10,25], by implementing a network-based robot control in a publisher–subscriber model using WebSocket [10]. The user interface of the software is shown in Figure 10.

Audio-Visual Streaming: The remote uses Gstreamer for video streaming over the network [59]. Our system can transmit real-time video feed captured at 60 frames per second with x264 encryption, which is sent with UDP to the local system. Data transmission could be initiated via a Windows command prompt. We set various attributes in order to establish the connections, including the local user’s IP address, port number, and the source of the camera device (i.e., ZED Mini Camera). Audio communication was implemented using WebRTC on Unity3D, similar to the local system.

## 6. Piton Control Methods

We implemented three control methods, HM—using a head-mounted display (HMD), HH—using an HMD and hand-held tracker, and FM—using an HMD and a foot-mounted tracker. The Appendix A show how each of the control methods are used to control Piton.

Previous research evaluated foot-controlled wearable robots, which utilized a linear control method to map foot movement to the robot’s movement [17,18]. In linear control, the robot’s workspace is measured and directly converted to a movement range for the user’s leg. Therefore, each point on the foot movement workspace is directly and linearly linked to each point in the robot’s physical workspace [17,18]. Despite the simplicity of linear control, an inevitable drawback of such a method is that it does not compensate for individual physiological differences among users’ bodies. For example, short users may not be able to extend their leg to completely cover the entire workspace of the robot while being seated.

In contrast to linear control, scalar control maps a user-defined workspace to the robot’s physical workspace, where the user-defined workspace does not necessarily match the dimensions of the robot’s physical workspace. Linear calibration and controls are widely used in telexistence systems [7,20], as they can easily be applied to map a user’s head rotations to those of the telexistence robot [7,17,20]. In scalar calibration, each point is calibrated to a different point of the other workspace to control the robot. The maximum and minimum movement can be scaled to a user’s defined body movement ranges. More generally, we can scale the movement ranges of the user’s hand or foot to compensate for users’ physiological differences. Accordingly, we utilized linear mapping for controlling the robot’s head rotations in all control methods, whereas we used scalar mapping for controlling positional movements (determined by the head, hand, or foot movements in each of the control method). To ensure proper execution of the controls, each user conducted an individual calibration of the intended control method before usage.

### 6.1. Calibration Procedures

Our calibration process should be conducted once for each user prior to using each of the control methods. The calibration is required as it ensures the produced robot movements cope with each user’s unique physiological movement ranges of their head, hand, and foot. Overall, we used the HMD for rotational movement in HM, HH, and FM control methods. The calibration process for rotational movement starts by instructing users to sit straight while wearing the HMD. Linear calibration of the rotational movement is initiated based on the first idle pose of the user’s head, which is set to zero in all rotational axes. Each rotational axis can enable a rotation of 140 degrees (70 degrees for each direction), corresponding to the user’s movement. Such movements directly control Piton’s head rotations within the same movement ranges, thereby directly controlling the three servomotor angles holding the ZED Mini Camera (pitch, yaw, and tilt).

While rotational movements are calibrated in the same way across all control methods, the calibration procedures for positional movements differ in each control method. The positional-movement calibration procedures are explained in the next subsections.

#### 6.1.1. HM

Positional movement in the HM is calibrated by instructing users to move their head to the lowest point near their left knee, then to move to the highest point on the top-back-right side (as shown in Figure 11a,b). Users were instructed to stretch their bodies as much as they comfortably can at each of the calibration points. Such calibration movements form a cube with coordinates at the bottom-front-left and top-back-right corners, where each point within such a workspace is scaled to a movement point in the robot’s IK [48] workspace to control the robot’s location.

#### 6.1.2. HH

The HH control method is calibrated by instructing users to hold the trackers with their dominant hand and moving the tracker to the left-bottom-front position, as high as their knee and as far as their hand extends (as shown in Figure 12a). Then, they are instructed to move their hand to the right-top-back, as high as their shoulder and as close as possible to their shoulder position (Figure 12b).

#### 6.1.3. FM

The calibration process starts by instructing users to attach the tracker on top of their shoes using velcro and to sit facing the calibration area, which is designated by the white square on the floor (as shown in Figure 13a,b). The white square is approx. 550 mm × 400 mm; such dimensions were selected to guide users during the calibration, as the calibration is scalar and adaptive to the user’s calibration procedure. Next, users are instructed to move their foot to the bottom-right and top-left edges of the white box as much as they comfortably can (as shown in Figure 13a). Vertical positional calibration is conducted by asking users to dorsiflex their foot and face upwards (bend the toes upwards while keeping their heels on the floor), as the difference between the natural foot posture on the floor and the dorsiflexed foot position (as shown in Figure 13b) determines the vertical workspace of the robot. Therefore, the vertical workspace was adaptive to each user’s maximum foot dorsiflexing angle. We chose this calibration procedure as we believe raising the foot in high positions may cause the users to become tired. The dorsiflexed pose enables users to have vertical movement while resting their foot on the floor (keeping the foot heel on the floor).

### 6.2. Using the Control Methods

To control Piton using each of the control methods (HM, HH, and FM), users can directly rotate Piton’s head by rotating their heads (while wearing the HMD). Positional movements are accomplished differently depending on the selected control method (as in Figure 14). The positional movements’ information, captured in real time through the HMD (HM) or the trackers (HH, FM), is fed to the coordinate system and set as an IK objective for the IK solver [48]. The received positional movements’ information is scaled to match the workspace of the robot’s movements. Next, the IK solver finds a solution that matches the set rotational positional movement objective in real time. Upon finding an IK solution that satisfies the set objectives, our system extracts the servomotor angle values of the provided solution and sends them to the robot’s control system over the network. Finally, the robot control system directly executes the servomotor angles on Piton through robot control software.

## 7. Evaluation

### 7.1. User Study Objectives

There are fundamental differences between human head movement and Piton’s movement. Compared to human head movement, Piton has a larger movement workspace due to its long snake-like form-factor. It can rotate on each axis at angles surpassing natural human movements. Such differences present a critical control issue, given that direct mapping between a user’s head and Piton cannot be easily achieved, both in terms of movements and orientations. These differences present challenges in achieving telexistence, especially as all surveyed telexistence systems use anthropomorphic robotic structures to directly match the user’s head movements with the robot’s movements.

Fundamental requirements of telexistence include low mental and physical loads [17], low motion sickness [20] and high body ownership [21] over the robot. Therefore, an essential objective of any control method of Piton is to have acceptable overall scores across these requirements. Therefore, we set the main objective of our evaluation to study the suitability of the three implemented control methods for use in telexistence, by focusing on the measurements of their mental and physical demands, motion sickness effects, and body ownership effects during various tasks. Accordingly, we used the NASA Task Load Index (NASA-TLX) [22] to measure users’ mental and physical demands. We used the Virtual Reality Motion Sickness Questionnaire (VRSQ) [23] to measure motion sickness effects. We used the Alpha IVBO questionnaire [24] to measure body ownership effects.

In addition to the mentioned requirements, we also explored various user impressions and opinions upon using these control methods using post-study questionnaires. The findings are significant in enabling us to understand the suitability and usability of the three implemented control methods across various contexts of use.

This study was conducted in Japan from September 2021 to May 2022 and was performed in accordance with the guidelines and procedures of the Office of Research Ethics of Waseda University.

#### 7.1.1. Participants

We recruited seventeen participants, fourteen males and three females, aged between 20–40 years (M: 26.56, SD: 5.92) from the university and outside the university. Participants came from various disciplines and backgrounds. All participants indicated that they had used VR at least once before and stated that they did not have significant eye-sight problems (six participants used eyeglasses with the HMD).

#### 7.1.2. Experimental Set-Up and Tasks

As shown in Figure 8, we mounted Piton on an aluminum frame, set the servomotor speeds to 17 rpm, and used the PID parameters explained in Section 5.3.1 for our user study. We chose this set-up and servomotor configurations for the following reasons: First, it is similar to wearing and using Piton while standing still in a remote environment. Second, to maintain safety while using Piton, especially since participants might rapidly move Piton as they try the three control methods. Third, this set-up enabled us to easily place a variety of equipment and objects that are needed for the different conditions of our experiment. Lastly, we chose these specific servomotor speeds, as higher speeds would cause high overshooting and camera-shaking, which would negatively affect our studied factors. Therefore, we instructed users to move their head and trackers at moderate speeds to match the robot’s speeds.

This experiment consists of three main tasks. They are explained below as follows:

Task 1 (mirroring) users have to look at a monitor that is placed in front of the robot to observe the robot’s body for one and a half minutes (as shown in Figure 15a,b). The monitor is equipped with a camera and acted as a mirror (similar to previous work [24], [60]). Mirroring is used to enhance virtual body ownership so that users can adapt to the three methods of controlling robot movement. This task is inspired by previous work on animal body ownership in VR [61].

Task 2 requires users to find and look at five randomly mentioned numbers and alphabetical letters from Uno Stacko [62]. The task is timed for a length of three minutes and involves approximately five trials (similar to previous work [63]). Uno Stacko is a block game where the goal is to match the color or number of the last block pulled and restack it on the top. In our study, we set Uno Stacko to show random numbers and letters with various colors on the front, left, right, and top sides (Figure 16a,b). This task was chosen to make users inspect Uno Stacko from multiple heights, directions, and orientations, resembling object inspection tasks [9].

Task 3 requires users to read a sentence that is printed and wrapped around a cardboard box, as shown in Figure 17a,b. The task is timed to finish in three minutes and involves approximately two trials. In each trial, users have to read a text from the left side to the right side of the box. We prepared six texts in total, and two texts are randomly used for each control method. This task was chosen to evaluate flexible head movement capabilities when scanning objects horizontally.

#### 7.1.3. Flow

The user study used a within-subject design and began with the participant completing a bibliographic questionnaire. Next, a researcher explained the experiment’s objectives, introduced Piton, and demonstrated its movement workspace. After that, the participant put on the HMD, and the researcher selected a random control method for the user. Each participant was given about one and a half minutes (similar to previous work [24], [60]) to familiarize themselves and observe how the IK robot model moves in response to the control method.

Next, Task 1 was conducted (one and a half minutes), followed by Task 2 (three minutes) and Task 3 (three minutes), where the order of performing task 2 and task 3 was randomized.

After performing Task 1, users completed the body ownership survey (based on Alpha IVBO [24]). Upon finishing Task 2 and 3, users completed the Virtual Reality Motion Sickness Questionnaire (VRSQ [23]) and mental and physical workload (NASA-TLX [22]). After that, the users were given a three-minute break. The same flow was repeated upon completing the surveys, albeit with a randomly selected control method. After completing all the conditions of the control methods, participants were given a survey to measure their overall preferences and impressions of the control methods.

### 7.2. Results and Analysis

#### 7.2.1. Quantitative Results and Analysis

##### Alpha IVBO

This questionnaire consists of three parts: acceptance, control, and change [24]. Acceptance corresponds to accepting the virtual body parts as one’s body parts, control corresponds to controlling the virtual body as one’s own body, and change corresponds to physiological or aesthetic changes felt during or after using a system. The questions were on a 7-point scale (1: strongly disagree to 7: strongly agree). The question results were grouped based on the three parts of Alpha IVBO (acceptance, control, and change), and then their mean values were calculated, as shown in Figure 18. A higher score indicates that the control methods were better in terms of users perceiving a virtual body as their own. The results of each of the Alpha IVBO parts are as follows.

Acceptance: the FM (M = 4.09, STD = 1.19) was rated lower than the HH (M = 4.44, STD = 1.00) and HH (M = 4.85, STD = 1.16).

Control: the FM (M = 4.90, STD = 0.77) was rated lower than the HH (M = 5.00, STD = 0.83) and HM (M = 5.16, STD = 0.87).

Change: the FM (M = 2.54, STD = 1.22) was rated lower than the HH (M = 3.08, STD = 0.96) and HM (M = 3.22, STD = 1.01).

A repeated-measures ANOVA was performed to confirm whether there was a difference between the robot control conditions in each subjective scale: the acceptance, control, and change. The repeated-measures ANOVA was selected since the experiment is a within-group design with more than two conditions and two independent variables.

The results of the repeated-measures ANOVA (as shown in Table 1) indicate that the acceptance and change are significantly different across the control methods, whereas control does not yield any significant effect.

As shown in Table 2, pairwise comparisons with a Bonferroni adjustment revealed that there was a statistically significant difference in the change between the HM and FM (*p* = 0.027) and the HH and FM (*p* = 0.021). The HM and HH did not yield significant results (*p* = 1.000).

##### NASA-TLX

The NASA-TLX consists of six subjective scales: mental demand, physical demand, temporal demand, performance, effort, and frustration. Each of these is divided into seven levels, from positive to negative evaluation with identical intervals [22], as shown in Figure 19.

Mental demand measures mental and perceptual activity requirements; physical demand assesses physical effort by the user during the task; temporal demand measures how pressured the users felt to complete the task; performance measures how successful users thought they were when completing the task; effort assesses how difficult it was to perform the task; and frustration measures the feeling of irritation, stress, or annoyance while completing the task. A lower score in each of the TLX subjective scale scores is considered better. Each of the subjective scale scores is as follows:

Mental demand: the HM (M = 2.06, STD = 1.25) was rated lower than the HH (M = 2.47, STD = 1.59) and FM (M = 2.88, STD = 1.50).

Physical demand: the HH (M = 2.29, STD = 1.26) was rated lower than the HM (mean = 2.59, STD = 1.54) and FM (M = 2.71, STD = 1.65).

Temporal demand: the HH (M = 2.59, STD = 1.42) was rated lower than the FM (M = 2.82, STD = 1.33) and HM (M = 2.88, STD = 1.54).

Performance: the HM (M = 1.88, STD = 0.86) was rated lower than the FM (M = 2.47, STD = 1.74) and HH (M = 2.53, STD = 1.46).

Effort: the HM (M = 2.53, STD = 1.50) was rated lower than the HH (M = 3.00, STD = 1.73) and FM (M = 3.29, STD = 1.79).

Frustration: the HM (M = 2.18, STD = 1.42) was rated lower than the HH (M = 2.18, STD = 1.47) and FM (M = 2.47, STD = 1.55).

A repeated-measures ANOVA was performed to confirm whether there was a difference between the control conditions in each subjective scale across the control method: mental demand, physical demand, temporal demand, performance, effort, and frustration. The repeated-measures ANOVA is selected since the experiment is a within-group design with more than two conditions and independent variables.

The repeated-measures ANOVA indicates that there are no differences for all the terms of the NASA-TLX across control methods, as shown in Table 3. The results show that the control methods produce a similar workload effect. These findings are further discussed within the Discussion section in Section 8.

##### VRSQ

The VRSQ is a motion sickness measurement index, specialized for virtual reality environments, that is widely used in various studies [23] and consists of an oculomotor component and a disorientation component, which is shown in Table 4. The oculomotor component covers general discomfort, fatigue, eye strain, and focus difficulty. The disorientation component accounts for headache, the fullness of the head, blurred vision, dizziness with eyes open, and vertigo.

The oculomotor score, disorientation, and VRSQ total score are calculated based on methods from a previous work [23], as shown in Figure 20. The computation of the oculomotor score is: ((attributes score)/12) × 100. The computation of the disorientation score is: ([attributes score]/15) × 100. The VRSQ total score is calculated as [Oculomotor score + Disorientation Score]/2. A lower VRSQ score is better as it indicates that the control method triggers fewer motion sickness effects.

The VRSQ’s lowest oculomotor score was for the FM (M = 12.25, STD = 9.37), followed by the HH (M = 15.20, STD = 18.69) and the HM (M = 20.59, STD = 20.86). Meanwhile, the lowest disorientation score was for the FM and HH (M = 3.14, STD = 4.78), followed by the HM (M = 5.49, STD = 5.89). Overall, the FM obtained the lowest VRSQ total score (M = 7.69, STD = 5.62), followed by the HH (M = 9.17, STD = 10.46) and HM (M = 13.04, STD = 11.64).

The Friedman test was performed to determine whether there was a significant difference in mean between the oculomotor component, disorientation component, and VRSQ total score of the three control methods. The Friedman test was selected because the experiment is a within-group design, and the data are not normally distributed. As shown in Table 5, the Friedman test results show there is a significant effect in the disorientation component across the control methods. However, the oculomotor component and VRSQ total do not yield any significant results.

Since there was a statistical significance in the VRSQ disorientation term, we conducted a post hoc analysis using Wilcoxon signed-rank tests with a Bonferroni correction to find where the differences are. As shown in Table 6, there is no significant difference in disorientation between the conditions (*p* is set to <0.0016). However, there are strong indications of a significant difference between the HM-HH and HM-FM (as shown in Table 6). We discuss the possible implications of this difference within Section 8.3.

#### 7.2.2. Qualitative Results

We asked users several questions to gauge the usability and impressions of the control methods (Q1: easiness, Q2: difficulty of looking at object, Q3: easiness of horizontal movement, Q4: easiness of vertical movement, Q5: rank the control methods from least to most liked, and Q6: rank the control methods in terms of subjective accuracy from most least accurate to most accurate). Q1–4 were asked individually after finishing each of the conditions (HM, HH, and FM) and consisted of a 6-point Likert scale (6 means best). Q5 and Q6 were asked after finishing all the conditions and consisted of ranking the control methods based on different subjective factors, where each control method should receive a unique rank.

In Q1, participants thought the easiest control method to move to the desired location (1: strongly disagree to 6: strongly agree) was the HM (M = 4.65, STD = 1.11), followed by the HH (M = 4.41, STD = 1.46), and FM (M = 3.94, STD = 1.43). In Q2, participants thought the easiest control method to look at specific objects (1: difficult to 6: easy) was the HM (mean = 4.59, STD = 1.12), followed by the HH (M = 4.47, STD = 1.42) and FM (M = 4.00, STD = 1.06).

In Q3, participants thought the easiest control method for the horizontal movement was the HM (M = 4.65, STD = 1.46), followed by the HH (M = 4.65, STD = 1.17) and FM (M = 4.18, STD = 1.29). In Q4, participants thought the easiest control method for the vertical movement was the HH (M = 5.18, STD = 1.07), followed by the HM (M = 4.24, STD = 1.60), and FM (M = 3.71, STD = 1.26).

In Q5, participants thought the most liked control method was the HM (M = 2.47, STD = 0.72) followed by the HH (M = 2.06, STD = 0.66) and FM (M = 1.47, STD = 0.80). Similarly, In Q6, participants thought the most accurate control method was the HM (M = 2.59, STD = 0.62) followed by the HH (M = 2.00, STD = 0.71) and FM (M = 1.41, STD = 0.71).

Users showed different responses to each question regarding usability (as shown in Figure 21). Overall, the users tended to select the HM for high scores over other control methods.

Each qualitative subjective result was further analyzed to reveal significant effects among the conditions. Therefore, we used the repeated-measures ANOVA to confirm whether there was a difference between the control conditions in each of the qualitative scales of Q1–Q6. If a change was found, we ran pairwise comparisons among the conditions to discover where the differences were exactly.

The results of the repeated-measures ANOVA (as shown in Table 7), indicates that the terms of Q4, Q5, and Q6 are significantly different across the control methods, whereas the terms of Q1–Q3 do not yield any significant effects.

Pairwise comparisons with the Bonferroni adjustment (as shown in Table 8) revealed that there was a statistically significant difference in the Q4 results between the HM and HH (*p* = 0.045), and the HH and FM (*p* < 0.001). The HM-FM (*p* = 1.000) did not yield significant results.

Pairwise comparisons with the Bonferroni adjustment (as shown in Table 9) revealed that there was a statistically significant difference in the Q5 results in the HM-FM (*p* < 0.025). Other conditions did not yield significant results.

As shown in Table 10, pairwise comparisons with the Bonferroni adjustment revealed that there was a statistically significant difference in terms of Q6 between the HM and FM (*p* < 0.001). Other conditions did not yield significant results.

To conclude, the HM control method was the most favored in terms of easiness of moving to the desired location, for looking at a specific object, moving horizontally, and was the most accurate in situating the robot at different postures during the task. The HM was also the second-most favorable in moving vertically during the tasks.

Meanwhile, the HH control method was the second-most preferred regarding its movement to the desired location, easiness of looking at a specific object, and accuracy of movement to locate the robot in various positions during the task; in addition, the HH control method is most favored in terms of vertical movement, and highly favored for horizontal movements.

Lastly, the FM control method is least preferred in terms of easiness of moving to the desired location, looking at a specific object, as well as moving horizontally and vertically. The FM was also least favored in terms of its overall accuracy in controlling Piton.

## 8. Discussion

In this section, we discuss the qualitative and quantitative results in light of our user study objectives. We conclude with discussing the suitability of each of Piton’s control methods within daily interaction contexts. The results are discussed in the following subsections.

### 8.1. Alpha IVBO Results

The IVBO-change results indicate that the HM and HH control methods scored significantly higher than the FM control method. This finding indicates that users felt a higher self-perception of the robot body through the HM and HH control methods than the FM. Having a higher self-perception (IVBO-change) contributes to better visual awareness of the surrounding environment and movement [64]. Although there are no significant effects between the conditions in IVBO-acceptance, the reported scores are high in all the control methods. Such results indicate that users felt self-attribution and body ownership of the robot’s body. Lastly, the reported IVBO-control results are high across all control methods, without significant effects between the conditions. The high IVBO-control indicates that the users felt a high agency while using Piton in all the control conditions and across the various tasks.

To conclude, all controls had high ratings for the body ownership in terms of IVBO. The only exception to this is the IVBO-change for the FM, which had a significantly lower rating than other control methods. We believe that this finding indicates a lower body ownership effect for the FM control method when controlling Piton.

### 8.2. TLX

The TLX results indicated that all control methods have low scores (below or equal to 3 on a 7-point scale) in each of the corresponding six subjective scales. The low mental and physical demand scores indicate that users did not feel mentally and physically exhausted in all the conditions of our study. The low scores on effort and performance indicate that the users could use each control method easily and successfully to accomplish the various tasks. The temporal term shows that users did not feel rushed to complete the task and generally had good pacing. Lastly, the low score in frustration indicates that users did not feel irritated or annoyed with the control methods. Our statistical significance testing did not reveal significant effects among the conditions in all the TLX terms. Therefore, we conclude that the various control methods had similar scores despite minor differences.

Overall, the low scores indicate that the control methods were not exhaustive or demanding across the various terms of the TLX. Nevertheless, it is critical to evaluate the control methods in lengthier user studies, which may result in different effects on users mental and physical efforts after extended usage. The results show that the HM, HH, and FM can be utilized for controlling Piton in various tasks, given that they are used for short periods.

### 8.3. VRSQ

The low VRSQ oculomotor score across the control method indicates that the users felt minimal motion sickness effects while using Piton (general discomfort, fatigue, eye strain, and focus difficulty). Similarly, the VRSQ disorientation score was low, thereby indicating that mental effects related to motion sickness (headache, fullness of head, blurred vision, dizziness, and vertigo) were not significant in all the control methods. Lastly, the VRSQ total score indicates that the control methods had minimal overall motion sickness effects.

Although our statistical analysis did not reveal specific significant effects between the conditions in the VRSQ disorientation score, there are strong indications that there was an effect. However, the effect was not strong enough to elicit significance. Such findings are especially apparent between the HM-HH and HM-FM (Table 6). We believe that there are stronger motion sickness symptoms from the HM than the HH and FM due to two main reasons. First, Piton is set to execute rotational and positional movements at a fixed speed during the evaluations. Although rotational movements are usually relatively short in duration, positional movements require the entire robot to take a new pose, which is more time-consuming. Accordingly, rapid movements by the user, or moving with varied acceleration/deceleration, may introduce an effect similar to latency in VR systems, which is a common cause of motion sickness [65]. Second, due to anatomical differences between the robot’s structure and the human head, the IK system occasionally provides solutions to positional movement objectives that are correct yet are executed with a slight mismatch to the user’s head location. For example, if the user quickly leans forward to inspect an object closely, the IK solver would produce a correct solution for the final pose. However, the produced trajectory is executed on the robot with minor changes that fit the robot structure and the IK model and objectives, thereby producing a slight trajectory mismatch. Eventually, this mismatch causes users to experience visually induced motion sickness. Second, due to anatomical differences between the robot’s structure and the human head, the IK system occasionally provides solutions to positional movement objectives that are correct yet are executed with a slight mismatch to the user’s head location. For example, if the user quickly leans forward to inspect an object closely, the IK solver would produce a correct solution for the final pose. However, the produced trajectory is executed on the robot with minor changes that fit the robot structure and the IK model and objectives, thereby producing a slight trajectory mismatch. Second, due to anatomical differences between the robot’s structure and the human head, the IK system occasionally provides solutions to positional movement objectives that are correct, yet executed with a slight mismatch to the user’s head location. For example, if the user quickly leans forward to inspect an object closely, the IK solver would produce a correct solution for the final pose. However, the produced trajectory is executed on the robot with minor changes that fit the robot structure, set objectives and the IK model structure, thereby producing a slight trajectory mismatch. Eventually, this mismatch causes users to experience visually induced motion sickness [66,67].

In comparison, the HH and FM do not require the user’s head to conduct positional controls, which we believe have contributed to their lower overall disorientation score. Overall, both discussed challenges can be addressed using the PID adjustments of the servomotors and adaptive control of Piton [68], which will significantly reduce the amount of delay between the user’s movements and the robot’s positional movements. Moreover, the IK system should further be enhanced to take into consideration the produced trajectory so that it completely follows the user’s head positional movements. For example, by assigning IK objectives to the middle joints, they can be precisely controlled according to the user’s head position. Furthermore, increasing the speed of the Piton control loop would also contribute to better and more responsive overall controls.

Overall, the results show that the control methods do not produce high motion sickness effects. Although there are significant differences that show the HM producing slightly higher motion sickness effects than other conditions, enhancements to the robot control loop and IK model can contribute to mitigating such issues. The HH and FM also have minimal motion sickness effects, which is encouraging to pursue extended deployments of such control methods further.

### 8.4. Qualitative Data

Overall, users thought the HM was easiest to utilize for moving Piton to a desired location and for looking at objects with relatively high accuracy. Participants also indicated that they thought the HM was the easiest to use, which we believe was mainly due to the control method’s simplicity and similarity with natural head movements.

Although binding Piton’s rotational and positional controls to the user’s head movements is intuitive for users, the limitations of human head movements cause a number of challenges. Participants indicated that some poses were difficult to accomplish, which was mainly due to limitations of the user’s natural head movement ranges. For example, it is difficult and tiring for users to rotate their heads up while extending their bodies forward. Similarly, it is difficult to access some locations in the calibrated space for positional controls, such as those directly near the user’s waist or thighs. Results from the questionnaire also show that users did not think the HM was best for vertical movements. Accordingly, our current calibration space is cubical in shape, and such a challenge can be addressed by creating a nonuniform calibration space that complies with the limitations of the human head movements. Therefore, despite its intuitiveness and ease of use, we believe that limitations caused by natural human head and neck movements affect the accessibility and usability of the workspace.

Users praised the HH control method for providing broad vertical and horizontal movement ranges that were easily accessible for rotational controls as well as vertical and horizontal positioning. In addition, the HH control method was highly rated by participants for the ease of movement to the desired location or for looking at an object. Our results also show that participants thought the HH control method was significantly more accurate in vertical movements than the HM and FM.

Although the rotational control was intuitive in the HH, positional control required coordinating the user’s hand movements with the head movements to situate the robot at different postures. Such coordination requires some practice, as users occasionally confuse head and hand movements, which leads to small control errors during the experiment. Similarly, visualizing the boundaries of the workspace, especially for horizontal movement, is an essential improvement to alert users upon reaching the extent of possible movement range. Accordingly, although the HH requires further head-hand movement coordination to control Piton, the HH has the highest subjective accuracy, especially for vertical movement, where these advantages overcome the movement limitations found in the HM.

The FM was least preferred by the users in terms of movement to a desired position and looking at an object, as well as for vertical and horizontal movement and subjective accuracy. The FM had low scores in terms of user preference, which indicates that users generally disliked the FM.

Further analysis of the qualitative results indicates that participants mainly attributed their dislike of the FM to the narrow workspace for positional movements using the foot. Overall, accessibility to various points in the control workspace is an issue, especially when their foot is very close or far away from them (as shown in Figure 13b). Such a limitation makes users unable to perform specific controls of Piton, such as raising Piton high at the forward-most or backward-most horizontal positions of Piton.

Participants also highlighted the difficulty of coordinating head and foot movements to control Piton, especially when compared to the HH. Despite the stated difficulties of the FM, users stated that it was not as tiring as the HM or HH. Users moved their foot within the tracking space while resting on their heels and dorsiflexed their toes up and down to control Piton. In contrast, the HM and HH require users to move their bodies or hand in order to control Piton, which could be more tiring during extensive positional movements or prolonged sessions. However, the NASA-TLX results did not show significant differences among the conditions.

Overall, the qualitative results show that users greatly favored the HM, the HH and then, the FM. We believe the main contributing factor for the user’s subjective preference of the HM over other methods is the familiarity of the control method, which resembled natural head movements. Their subjective accuracy scores show that they felt the HH was more accurate in controlling Piton, then the HM, and lastly, the FM. The HH control method’s main advantage is the accessibility and flexibility in horizontal and vertical movements that surpass the HM and FM, which are mainly limited due to the user’s natural head and foot movement limitations.

### 8.5. The Suitability of the Control Methods in Various Contexts of Use

The HM control method had a low TLX score, which showed low mental and physical demand. At the same time, a low overall VRSQ score indicated only a slight motion sickness effect. The HM control method also had higher body ownership, which shows a high self-perception of the robot as the user’s own body. The qualitative analysis also indicates that users favor the HM over other methods due to its intuitiveness and praised it for its relatively high accuracy. Such a rating was mainly due to the HM resembling human head movements, making it familiar and easy to use. The HM also had minimal motion sickness effects. Therefore, we believe the HM control method is suitable for tasks that do not require high accuracy but require ease of use and comfortable control methods that any person can use. A typical example of this is daily usage tasks, such as companionship during travel, hiking, or shopping. Additional tasks also include remote assistance and guidance at home, such as teaching users how to cook or to set-up and operate a device.

The HH control method had low mental and physical demand, low motion sickness, and higher body ownership, which is similar to the HM. Users praised the HH for its relatively high accuracy, especially for vertical movement. Therefore, we believe the HH control method is suitable for tasks that require high accuracy in positional movements. For example, industrial tasks that require inspection of tools or objects from multiple angles and distances.

Lastly, the FM control method had a low score for both mental-physical demand and motion sickness. The FM control method had a lower IVBO-change score and higher IVBO-acceptance and IVBO-control scores, meaning this control method had fair body ownership scores. The qualitative results show that users least-liked the FM because other control methods were more accurate. However, the FM can potentially be effective for longer usage sessions, as users can relax their foot on the floor while also being able to control Piton. Therefore, the FM control method can be utilized for daily life tasks or industrial tasks that do not require high accuracy but require prolonged usage sessions.

Overall, we believe that each control method has different advantages and disadvantages in various contexts, especially since the control method should be designed based on task interaction requirements [69]. Our evaluation of the control methods provides various insights for the usability of the control methods. We also believe the sample size is suitable for studying Piton’s control methods and elicits various insights about its usability within various contexts. However, we believe that a larger and more varied sample size may yield extended results, especially about the suitability of these control methods within various tasks within daily or industrial contexts.

## 9. Conclusions and Future Work

This paper presents Piton, a novel, wearable, snake-like telexistence robot. Piton can be used in various contexts, whether for leisure applications or industrial and professional contexts. We discuss Piton’s implementation specifications and explain three control methods that we used for controlling Piton.

Although all the control methods generally had high NASA-TLX scores, high body ownership, and low motion sickness effects, there exists a number of differences that distinguish the control methods, especially in terms of qualitative results. The HM control method has the highest body ownership results and was most favored by participants due to its intuitiveness, as it resembles natural human head motion. Therefore, the HM is best deployed for tasks that require basic movements, such as those within daily usage contexts. The HH control method has the highest perceived accuracy, since users could easily position Piton in various locations using the hand-held tracker. Therefore, we believe it is best suited for tasks requiring high accuracy, such as industrial inspection tasks. The FM was least-liked by the participants, who also thought it was imprecise for the evaluated tasks. However, this control method enables users to relax their foot on the floor while controlling the robot, which could be suitable for longer usage sessions.

Although our user study yielded various insights for the usability of the implemented control methods for telexistence, it is essential for future work to explore additional control methods using other modalities of control. Moreover, expanding the user study with a larger and more diverse sample size would enable us to gather deeper insights, especially with users who are not familiar with VR or with those coming from industrial backgrounds. Piton should also be improved, especially by increasing its DoFs and further stabilizing its camera, which would increase its movement flexibility, increasing its speed and overall user experience.

An important finding of our work is that our results show that a control method that does not resemble natural human head movements, such as HH, can be superior for robot control and does not jeopardize essential telexistence requirements (e.g., relatively high body ownership, low motion sickness, and TLX scores). This aspect is encouraging for further exploration of future control methods that can both provide an adequate balance between telexistence experience and task efficiency and may include other control modalities for accurate robot controls.

Most importantly, Piton shows that robots with different anatomical designs than humans can be used for telexistence. Although Piton has a different anatomical structure and workspace than a human neck, our developed control methods of the kinematic model and IK solver could yield a suitable telexistence experience. Such findings pave the way for further research to explore other nonhuman form-factors for telexistence. Such form-factors can provide various benefits beyond human-mimetic telexistence robots, such as higher accuracy, a larger movement workspace, or more interaction capabilities.

The design and evaluation of Piton revealed several research opportunities and challenges to build on our presented robot and research results. Based on our design and evaluation of Piton, we discuss several design improvements and future research directions that are essential to build on our presented efforts:

Motion Sickness: In addition to the enhancements discussed in Section 8.3, we believe camera vibrations and shaking, which are caused by the servomotors or rapid movements, contributed to motion sickness. Such challenges can be addressed in a variety of improvements. Motor control optimizations (e.g., PID control) can stabilize and smooth out the robot’s movements, as well as decrease overshooting during faster movements. In addition, adding elastic and soft materials to the camera holder can absorb vibrations caused by the servomotors, which in turn, can significantly reduce shaking during movements. Some of these improvements are widely used in drones to reduce the shaking caused by atmospheric factors and high-speed motors [70,71].

Our VRSQ results showed that most users had minimal motion sickness while using Piton. However, participants of our user study used Piton in short bursts (3–5 min), and we allocated sufficient resting time between the tasks. Therefore, it is important to evaluate the usability of Piton for prolonged sessions, especially as our initial tests revealed that using the Piton for long periods induces high motion sickness effects.

Robot Structure: Inspecting an object from various distances and angles is an important capability of Piton. However, the current implementation limits horizontal movement of the robot as such movement is conducted using two servomotors (first and third servomotors). Therefore, in order to extend the movement range of Piton, more DoFs are needed. Therefore, adding more servomotors next to the third servomotor for backward and forward positional movements can extend the range of horizontal motions.

Safety and Hazards: Although our robot is generally underpowered, we used a position-based control method to control the robot. The robot’s sudden and quick movements may cause hazards to the users, especially near its base (where we use stronger servomotors). Therefore, we intend to utilize torque-based position controls or impedance controls, which can enable the wearer to easily push away the robot in case of emergencies without much effort.

Supplementary Interaction Methods: Further research should explore using additional interaction methods to supplement HMD-based controls, such as using eye-gaze [72,73], electromyography [74,75], or a hand-mounted exoskeleton [76]. We believe these additional interaction methods are especially needed to extend controllability of Piton, especially for positional movements and postures that are uncomfortable for users to execute using our implemented control methods (e.g., positions too close/low to the user). Integrating these control methods may also contribute to higher accuracy and comfortable usage during extended sessions, especially since physical controls utilizing the user’s head, hands, or legs are tiring after prolonged use. However, the effects of using supplementary control methods should also be evaluated within the context of telexistence.

Wearability Evaluation: Although our work focuses on Piton control methods, its wearability within daily and industrial contexts presents numerous challenges. First, since the wearable robot is kinematically dependent on the user wearing it (at the remote site), the robot may be moved involuntarily by the user wearing Piton, which may induce high motion sickness. One method for addressing this challenge is using inertia measurement units (IMUs) to moderate the effects of the surrogate’s movements on robot movements.

Second, sudden movements of the robot at high speeds cause the backpack rack to rapidly shake. Therefore, the backpack rack and the robot base should be further strengthened and stabilized. Lastly, similar to other innovative robot form-factors [76], Piton presents interaction potentials that are not explored in prior works. Therefore, we believe that workshops should be held with both professional and casual users to explore future potential application domains of Piton. The outputs of such workshops would deepen our understanding of the requirements and expectations of using robots such as Piton.

Extended Evaluations and Task Domain Investigation: Our main evaluation results showed the advantages and disadvantages of each of Piton’s control methods within the context of telexistence. Such results are essential to pave the way for extended evaluations that focus on larger and varied user groups, as well as deeper usability studies of daily life or industrial tasks. Accordingly, future evaluations should focus on large-scale user studies involving larger and varied user groups. Moreover, the interactions that occur between the controller and surrogate users, and between those mentioned users and the remote environment, should be studied in the context of using Piton as the sole communication medium to fulfill different tasks. Another important direction of evaluations is to have deeper explorations of the deployment domains of Piton within daily usage and industrial contexts. To fulfill this objective, focus groups and workshops should be conducted [9,44,45], where such evaluations explicitly study the requirements, expectations, and deployment tasks and contexts of robotic systems. Overall, such findings are critical for the adoption and deployment of Piton within real-world daily use and industrial contexts.

In summary, our design and evaluation of Piton revealed several opportunities and challenges. Various mechanical and technical enhancements are required, such as camera stabilization and robot structure enhancements, that may directly contribute to a better user experience. Utilizing adaptive control strategies to select the appropriate control method based on the needed task is an important direction when deploying robots such as Piton. Moreover, other control modalities should also be explored, especially as physical controls are often tiring for users during extended usage sessions. Further research directions should also focus on exploring Piton as a robotic appendage for use within both daily and industrial contexts, which will pave the way for its effective deployment within such contexts.

## Figures and Tables

**Figure 1 sensors-22-08574-f001:**
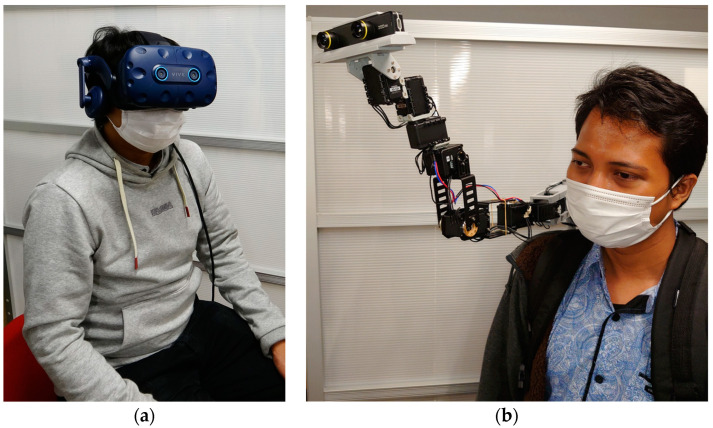
Design concept of Piton. (**a**) At the local site, the user uses the HMD to interact with Piton at the remote site. (**b**) At the remote site, a surrogate user wears Piton.

**Figure 2 sensors-22-08574-f002:**
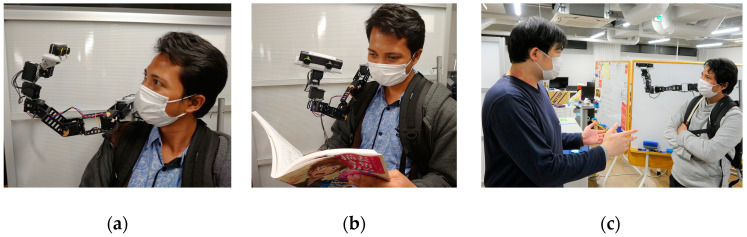
Piton can be used in everyday usage contexts. For example, Piton can be used to (**a**) interact with the surrogate, (**b**) interact with the remote environment, sharing various experiences with remote users or checking merchandise, or (**c**) enjoy the outdoor scenery or social activities.

**Figure 3 sensors-22-08574-f003:**
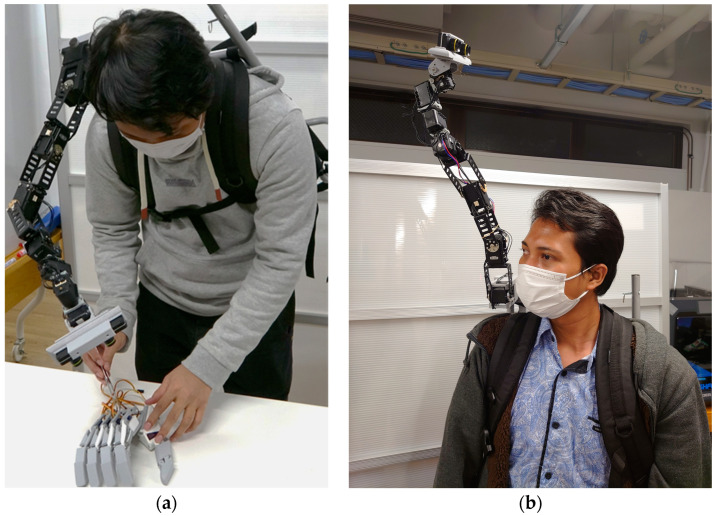
Piton can be used for industrial tasks. (**a**) Piton can support remote knowledge transfer and training, such as instructing remote users during assembly or machinery operation tasks. (**b**) The flexibility of Piton can be used for inspecting objects or environments, such as by extending around or above the surrogate user.

**Figure 4 sensors-22-08574-f004:**
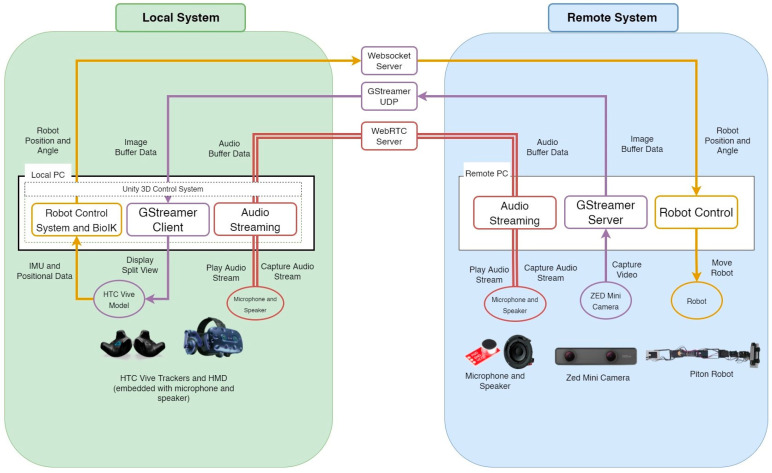
This diagram shows the overall architecture of our system. The arrows indicate the data flow between the various components, coded in three colors: red for auditory communication, purple for stereoscopic video streaming, and yellow for robot control.

**Figure 5 sensors-22-08574-f005:**
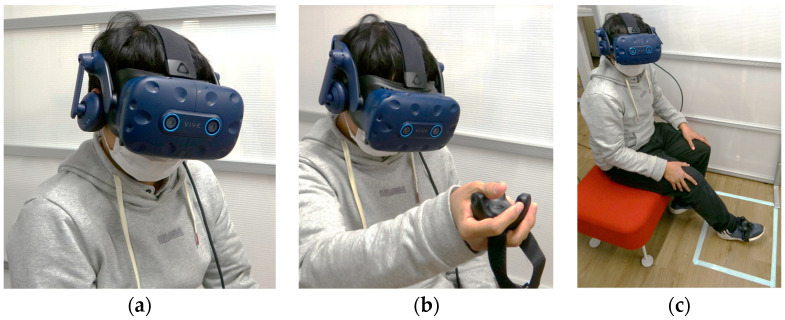
Control methods: (**a**) HMD to control the position and orientation of Piton; (**b**) HMD to control Piton’s rotation and a hand-held tracker to control its position; (**c**) HMD to control Piton’s rotation and a foot-mounted tracker to control its position.

**Figure 6 sensors-22-08574-f006:**
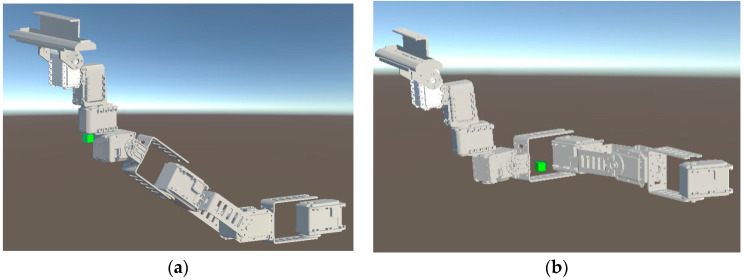
Visualization of the IK system and robot model, with the green cube presenting the target objective for the positional movement: (**a**) The robot moves to top position; (**b**) The robot moves to the right position.

**Figure 7 sensors-22-08574-f007:**
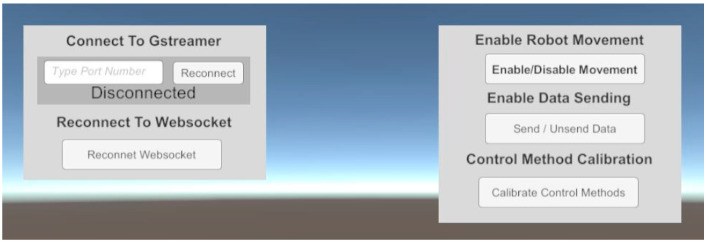
The graphical user interface of our system connects with Gstreamer and WebSocket server (robot control software), allows enabling/disabling the robot’s movements or sending data, and for starting the calibration process of the control methods.

**Figure 8 sensors-22-08574-f008:**
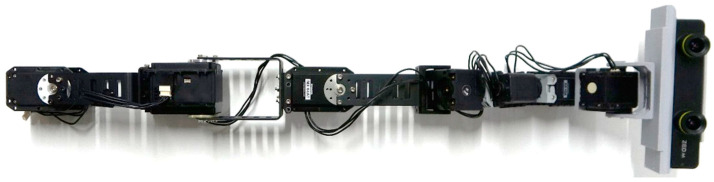
Piton robot structure. The robot is composed of eight servomotors interlinked using aluminum and PLA brackets. The end-effectors comprise a PLA ZED camera holder.

**Figure 9 sensors-22-08574-f009:**
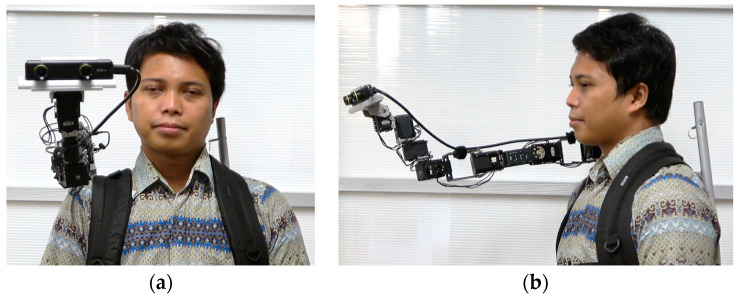
The robot is mounted on a backpack rack. (**a**) Front view; (**b**) side view.

**Figure 10 sensors-22-08574-f010:**
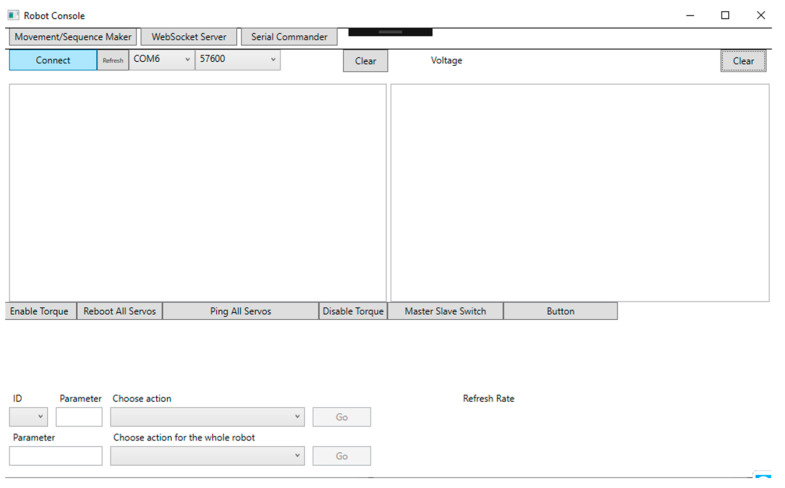
The robot control software’s UI enables controlling the robot through WebSocket.

**Figure 11 sensors-22-08574-f011:**
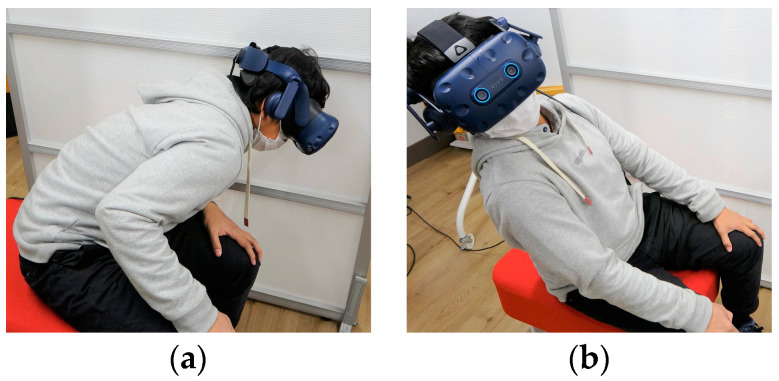
HM control calibration procedure of positional movement. The user moves their head to the corners of the control space, as shown in (**a**,**b**), thereby forming a tracking area that maps the user’s head position to the robot’s neck position.

**Figure 12 sensors-22-08574-f012:**
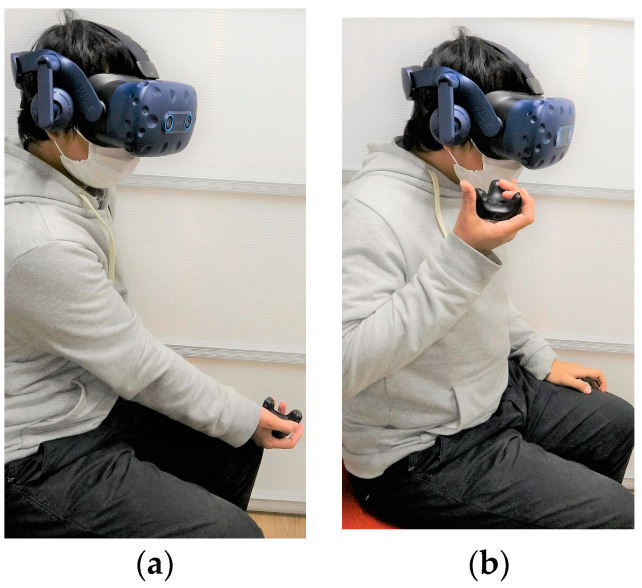
HH control calibration procedure. The user moves their hand to the corners of the control space, as shown in (**a**,**b**), thereby forming a tracking area that maps the user’s hand position to the robot’s neck position.

**Figure 13 sensors-22-08574-f013:**
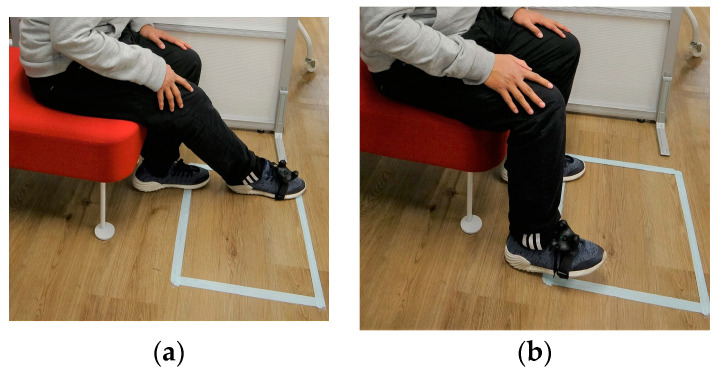
FM control calibration procedure. The user moves his foot to the corners of the control space, as shown in (**a**,**b**), thereby forming a tracking area that maps the user’s foot position to the robot’s neck position. Dorsiflexing their foot upwards calibrates the vertical positional movement, enabling them to move Piton within the calibrated workspace.

**Figure 14 sensors-22-08574-f014:**
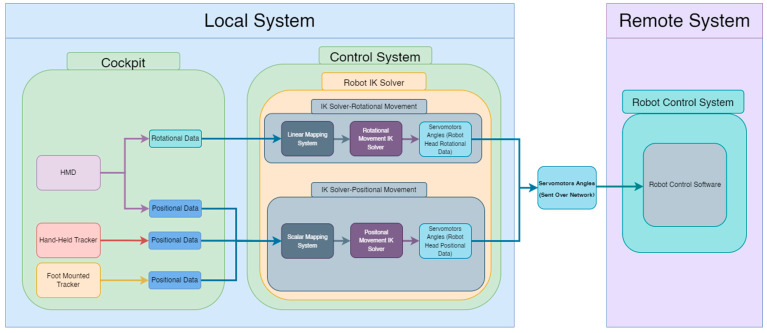
This diagram illustrates how each data point is captured from the HMD and trackers, and then processed to produce servomotor angles through our control system at the local site. The servomotor angles are then sent to the remote site where they are executed by the robot control software.

**Figure 15 sensors-22-08574-f015:**
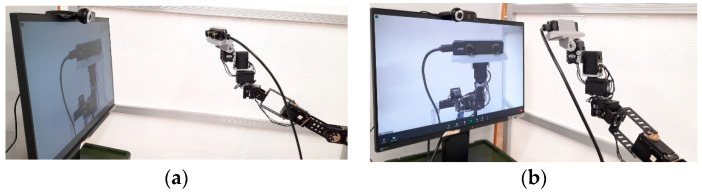
Task 1 (mirroring): (**a**) a monitor with a web camera is used for mirroring task; (**b**) the user can observe the robot’s movements by looking at the screen (similar to a mirror).

**Figure 16 sensors-22-08574-f016:**
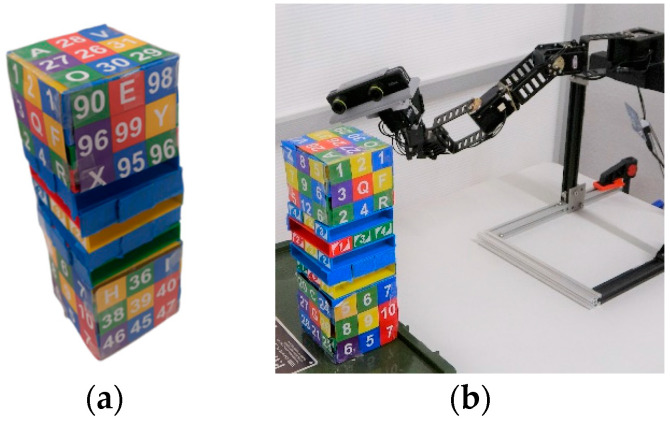
Task 2 (finding numbers and letters): (**a**) Uno Stacko block game with randomly set numbers and letters; (**b**) A user moving Piton to find specifically colored numbers during task 2.

**Figure 17 sensors-22-08574-f017:**
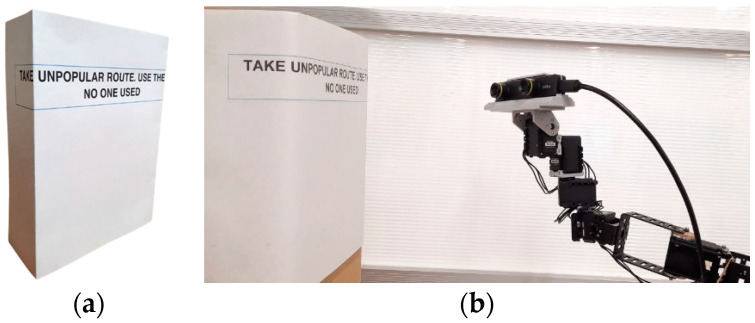
Task 3 is text reading: (**a**) the text is printed and wrapped around a box, and (**b**) users have to control Piton to look around the box edges to read the text.

**Figure 18 sensors-22-08574-f018:**
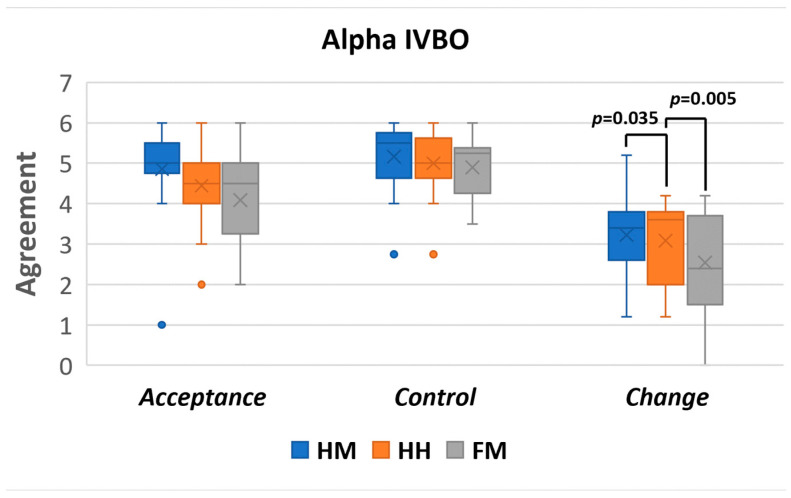
Alpha IVBO questionnaire results: acceptance, change, control.

**Figure 19 sensors-22-08574-f019:**
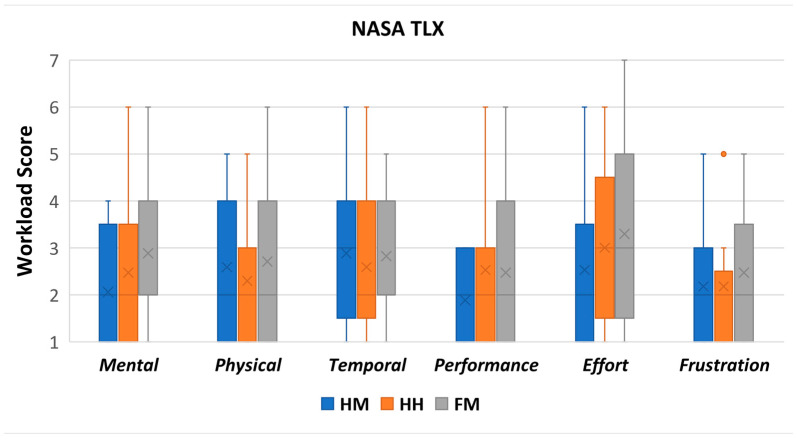
NASA-TLX results of the 17 participants.

**Figure 20 sensors-22-08574-f020:**
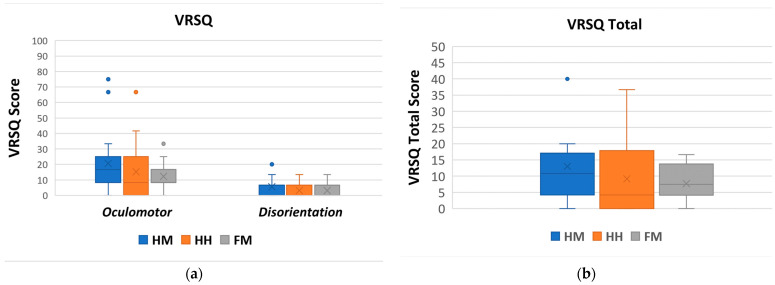
VR Sickness Questionnaire: (**a**) results of oculomotor score and disorientation score; (**b**) results of VRSQ total score.

**Figure 21 sensors-22-08574-f021:**
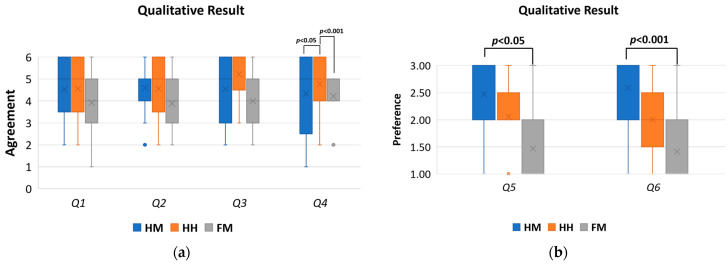
Results of the poststudy questionnaires: (**a**) results of Q1–4; (**b**) results of ranking questions (Q5, Q6).

**Table 1 sensors-22-08574-t001:** The results of repeated-measures ANOVA.

Source	SS	df	Mean Square	F	*p*-Value
Acceptance	4.980	1.570	3.173	4.190	0.035 *
Control	0.605	1.473	0.411	1.054	0.344
Change	4.411	1.737	2.540	6.846	0.005 *

***** The significant value is <0.05.

**Table 2 sensors-22-08574-t002:** IVBO-change results of pairwise comparisons with Bonferroni adjustment.

Control Method	Mean Differences (I–J)	Std. Error	*p*-Value
HM-HH	0.141	0.175	1.000
HM-FM	0.682	0.229	0.027 *
HH-FM	0.541	0.175	0.021 *

* The significant value is <0.05.

**Table 3 sensors-22-08574-t003:** The results of repeated-measures ANOVA.

Source	SS	df	Mean Square	F	*p*-Value
Mental Demand	5.765	1.703	3.385	2.412	0.115
Physical Demand	1.529	1.398	1.094	0.512	0.542
Temporal Demand	0.824	1.935	0.426	0.642	0.528
Performance	4.353	1.871	2.327	2.403	0.111
Effort	5.059	1.970	2.567	2.932	0.69
Frustration	0.980	1.916	0.512	0.567	0.566

**Table 4 sensors-22-08574-t004:** Motion sickness component mean scores and standard deviation values for each control method.

Method		General Discomfort	Fatigue	Eye Strain	Focus Difficulty	Headache	Fullness of the Head	Blurred Vision	Dizzy	Vertigo
HM	M	0.53	0.65	0.88	0.41	0.35	0.18	0.24	0.06	0.00
	STD	0.51	0.70	1.05	0.87	0.49	0.39	0.44	0.24	0.00
HH	M	0.29	0.53	0.65	0.35	0.00	0.12	0.29	0.06	0.00
	STD	0.59	0.62	1.00	0.61	0.00	0.33	0.59	0.24	0.00
FM	M	0.41	0.41	0.47	0.18	0.06	0.12	0.18	0.12	0.00
	STD	0.51	0.62	0.62	0.39	0.24	0.33	0.39	0.33	0.00

**Table 5 sensors-22-08574-t005:** The results of Friedman test of VRSQ.

Source	N	Chi-Square	df	*p*-Value
Oculomotor	17	2.577	2	0.276
Disorientation	17	8.000	2	0.018 *
VRSQ Total	17	4.230	2	0.121

* The significant value is <0.05.

**Table 6 sensors-22-08574-t006:** Wilcoxon signed-rank test results on the VRSQ disorientation term.

	HM-HH	HM-FM	HH-FM
Z	−2.251	−1.732	0.000
*p*-Value	0.024	0.083	1.000

**Table 7 sensors-22-08574-t007:** The results of repeated-measures ANOVA on the results of Q4–6.

Source	SS	df	Mean Square	F	*p*-Value
Q4	23.111	1.643	14.068	7.158	0.005*
Q5	8.588	1.903	4.514	5.407	0.011*
Q6	12.704	1.788	7.107	7.208	0.004*

* The significant value is <0.05.

**Table 8 sensors-22-08574-t008:** Results of pairwise comparisons with Bonferroni adjustment of Q4 results.

Control Method	Mean Differences (I–J)	Std. Error	*p*-Value
HM-HH	−1.111	0.411	0.045 *
HM-FM	0.444	0.506	1.000
HH-FM	1.556	0.336	<0.001 *

* The significant value is <0.05.

**Table 9 sensors-22-08574-t009:** Results of pairwise comparisons with Bonferroni adjustment of Q5 results.

Control Method	Mean Differences (I–J)	Std. Error	*p*-Value
HM-HH	−0.412	0.272	0.449
HM-FM	−1.000	0.332	0.025 *
HH-FM	−0.588	0.310	0.228

* The significant value is <0.05.

**Table 10 sensors-22-08574-t010:** Pairwise comparisons with Bonferroni adjustment of Q6 results.

Control Method	Mean Differences (I–J)	Std. Error	*p*-Value
HM-HH	−0.778	0.319	0.078
HM-FM	−1.167	0.259	<0.001 *
HH-FM	−0.389	0.354	0.861

* The significant value is <0.05.

## Data Availability

All data collected during this research is presented in full in this manuscript.

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
