# Peer review of "Piton: Investigating the Controllability of a Wearable Telexistence Robot"

_sensors, 2022, doi:10.3390/s22218574_

Round 1

Reviewer 1 Report

The authors propose interesting studies of a Wearable Telexistence Robot. However, there are some issues that need to be improved.

-          The authors have declared the novelty of their proposed robot. But it is still hard to understand and distinguished their novelty compared to other studies

What is your novelty compared to other existing research studies ?

 -          Do you develop your own tracking system algorithm for the HMD and tracking system ?

 -          For the feedback control, the authors mentioned PID compensator.  Can you measure the error during the operation of your robot system.

 -          For the calibration using three methods (HM, HH, and FM), if a new user wants to use your system, does the calibration always needed for a new user?

 -          The authors implemented a publisher-subscriber using WebSocket for the network communication control. Can you measure/estimate the delay from a user (local site) to the pyton robot?

 -          The authors presented a simple application for their robot operation. Can you provide more complex application for your robot Wearable Telexistence Robot.

 -           For the three control methods (HM, HH, and FM), it will be better if the author can provide a block diagram consisting the feedback control parameters such as control, sensors (measured parameters), and desired parameters

 Minor comments:

-          Page 17, (as shown in Table 1Error! Reference source not found.),

-          The table format should follow the journal template

Author Response

We revised the Manuscript in order to improve its overall quality according to the reviewers’ suggestions and feedback. First, we revised the introduction to include a “contribution subsection” as suggested by the reviewers, which summarizes and clarifies the contributions of our work. We also revised the conclusion and merged it with Future works based on R2, forming a new section called “Conclusion and Future Work” which highlights the main conclusion of our work and future research direction to build on our efforts. To clarify our control system algorithm, We added a new block diagram to explain how data is gathered from the input devices, processed, and used to generate motion on the robot for each control method. In addition, we fixed style format such as Table to the MDPI-Sensor proper format and implemented many small fixes (typos, missing diagram links, misaligned text…etc). Please see the attachment for more detail.

Reviewer 2 Report

Reviewer: 1

Dear Editor in Chief

It proposed a snake-shaped wearable telexistence  robot, called Piton, that can be remotely used for a variety of collaborative applications. It explained the implementation of Piton, its control architecture, and discuss how Piton can be deployed in a variety of contexts.

After reviewing this paper, main comments are listed as:

1-      Section “8. Design Improvements and Future Work” must be deleted and it should be represented at conclusion section. In the conclusion section, the future work is not explained.

2-     Separate sections for "Main contribution" is required.

3-     Paper is very high volume of contexts and figure and It should be brief at context and figures.

4-     Table 1 and Table 4 are not presented in suitable format and margin; It should be revised.

5-     This is a research paper, and it is not a technical report so “Table 7. The list of questions was used for our qualitative analysis” should be eliminated.

6-     Figure 14.  and Figure 6 and Figure 2 have a very high context, and it is not usual for research paper; it should be brief in context.

7-     In 3. Design Concept, author explained two sections also it is kind of technical report and author should change this section based on research paper.

8-     Several recent works should consider a wearable robot and control strategies was proposed for and author must review advantageous and disadvantageous of these approaches. I suggest to author to review following paper:

1-     Real-time implementation of a super twisting control algorithm for an upper limb wearable robot, Mechatronics 84, 102808.

2-      Design of a Wearable Bilateral Exoskeleton for Arm Stroke Treatment in a Home Environment- 2021 27th International 

3-     An Industrial Robot-Based Rehabilitation System for Bilateral Exercises, in IEEE Access, vol. 7, pp. 151282-151294, 2019.

Author Response

(The authors gave the same response as above.)

Reviewer 3 Report

The paper describes the implementation and some initial testing of a small snake robot mounting a vision system and used for tele-existence tasks. The shape of the robot is mainly determined by the aim of improving space exploration through vision. The telecontrol is obtained with 3 different interfaces, tested on 17 volunteers, on 3 various tasks.

According to the authors: “Our paper focuses on presenting Piton’s novel form factor, its potential application domains extracted from related literatures and the covid-19 global pandemic restrictions on social interactions, and an evaluation that focuses on the controllability of Piton within the context of Telexistence.”

The novel form factor is not so new; a previous paper (reference 30) has already discussed a similar snake robot equipped with haptic devices.

Anyhow, the construction of the robot is quite simple. Since many years it is common for university students to make rapid prototyping of complex robots using a toolbox (for instance bioloid) to build the mechanical structure and to move many motors together. The piton robot could be easily built with such tools. Moreover, recent software components as the mentioned BioIK are effective in designing the controller. The large use of off-the-shelf tools is of curse acceptable.

In conclusion, building the robot itself is an exercise; the novelty of the paper is in testing different user interfaces in various tasks. This evaluation has some statistical results, is commented in the paper, but given the small number of collected data is preliminary.

Another point to clarify is about terminology. According to the book “Wearable Robots: Biomechatronic Exoskeletons”, by J.L. Pons, ISBN: 978-0-470-51294-4, 2008, “A wearable robot is a mechatronic system that is designed around the shape and function of the human body, with segments and joints corresponding to those of the person it is externally coupled with.”

Wearable is not the same as portable. Wearable follows the shape and function of the human body. In this paper the robot is portable, and helps the user in moving the camera around. The term “wearable” is applied to the snake robot because it is used to extend the vision capabilities of the human carrying it. However, it does not improve the human vision ability, but only offers video images. As observed by the authors, the difference in the structure of the human head and the snake robot is responsible of some problems of visual sickness because of the mismatch between the two different trajectories.

The main drawback of the paper is that all the results are obtained after using the robot for a few minutes, a situation different from real operating conditions in working environments, or in disaster recovery, where the activity lasts for hours.

In conclusion, the authors are invited:

-       to improve the Introduction to explain what is new in the paper,

-       to discuss about the extended use of the term wearable and to add warnings about the problems it can cause, and if possible

-       to report preliminary results of longer and more complicated tasks (may be already available from a few coworkers)

-       to improve the style of the paper.

Finally the authors should explain how the ethical committee of their institution has managed the use of human people in the experimentation.

Author Response

(The authors gave the same response as above.)

Round 2

Reviewer 1 Report

The authors have improved their work significantly.

Reviewer 3 Report

The paper describes the implementation and the initial testing of a small snake robot used for tele-existence tasks. The telecontrol is obtained with 3 different interfaces, tested on 17 volunteers, on 3 various tasks.

The authors in the revised paper have addressed the main points raised during the first review.

The target of the paper is testing 3 different user interfaces, in various tasks. This is clearer now.

About the simplicity of the robot they say that this guarantees reproducibility. This is true; however, this implies that the control is not intuitive and the used has to learn it.

The discussion about extending the term “wearable “ to Piton has been added.

The authors recognize that the evaluation of the robot in real tasks should be done in future work.

In conclusion, the paper has a more balanced presentation of the results and the claims are better motivated.